# Ground Deformation and Permafrost Degradation in the Source Region of the Yellow River, in the Northeast of the Qinghai-Tibet Plateau

**Chengye Li** [1], **Lin Zhao** [1,2,*], **Lingxiao Wang** [1], **Shibo Liu** [1], **Huayun Zhou** [2,3], **Zhibin Li** [1], **Guangyue Liu** [2,3], **Erji Du** [2,3], **Defu Zou** [2] and **Yingxu Hou** [1]

1    School of Geographical Sciences, Nanjing University of Information Science & Technology (NUIST), Nanjing 210044, China; lichengye@nuist.edu.cn (C.L.); lx.wang@nuist.edu.cn (L.W.); liushibo18@mails.ucas.ac.cn (S.L.); zhibinli@nuist.edu.cn (Z.L.); houyingxu@nuist.edu.cn (Y.H.)

2    Cryosphere Research Station on the Qinghai-Tibet Plateau, State Key Laboratory of Cryosphere Science, Northwest Institute of Eco-Environment and Resources, Chinese Academy of Sciences (CAS), Lanzhou 730000, China; zhouhuayun18@mails.ucas.ac.cn (H.Z.); liuguangyue@lzb.ac.cn (G.L.); duerji@lzb.ac.cn (E.D.); defuzou@lzb.ac.cn (D.Z.)

3    University of Chinese Academy of Sciences, Beijing 100864, China

*    Correspondence: Correspondence: lzhao@nuist.edu.cn

**Abstract:** The source region of the Yellow River (SRYR) is situated on the permafrost boundary in the northeast of the Qinghai-Tibet Plateau (QTP), which is an area highly sensitive to climate change. As a result of increasing global temperatures, the permafrost in this region has undergone significant degradation. In this study, we utilized Sentinel-1 to obtain ground surface deformation data in the SRYR from June 2017 to January 2022. We then analyzed the differences in terrain deformation under various environmental conditions. Our findings indicated an overall subsidence trend in the SRYR, with a long-term deformation velocity of −4.2 mm/a and seasonal deformation of 8.85 mm. Furthermore, the results showed that terrain deformation varied considerably from region to region, and that the Huanghe' yan sub-basin with the highest permafrost coverage among all sub-basins significantly higher subsidence rates than other regions. Topography strongly influenced ground surface deformation, with flat slopes exhibiting much higher subsidence rates and seasonal deformation. Moreover, the ground temperature and ground ice richness played a certain role in the deformation pattern. This study also analyzed regional deformation details from eight boreholes and one profile line covering different surface conditions, revealing the potential for refining the permafrost boundary. Overall, the results of this study provide valuable insights into the evolution of permafrost in the SRYR region.

**Keywords:** permafrost; ground surface deformation; SBAS-InSAR; source region of the Yellow River (SRYR)





## 1. Introduction

Permafrost degradation is occurring at an alarming rate due to global warming [1,2]. The Qinghai-Tibet Plateau (QTP), with an average elevation of over 4000 m, has the world's largest permafrost distribution at low and middle latitudes, covering about $1.06 \times 10^6$ km² [3]. The QTP is the source of numerous rivers. As an important part of China's Water Towers, the source region of the Yellow River (SRYR) is particularly important, providing almost 38% of the total annual runoff [4]. The SRYR is situated at the northeastern border of the Tibetan Plateau and is influenced by the East Asian monsoon and the western Pacific paramount, making it highly sensitive to climate change [5–12]. It is also located in the interface area between seasonally frozen ground and permafrost boundaries [3,13,14]. The permafrost in the transition zone has undergone significant degradation, resulting in increased ground temperatures, thinning of permafrost thickness, development of taliks

and the disappearance of permafrost islands and deeply buried permafrost [6]. The active layer thickness in the SRYR is increasing at a rate of 0.55 cm/a, and from 2003 to 2019, approximately 4.82% of the permafrost degraded to seasonally frozen ground [15]. These changes have led to ecological degradation, such as soil erosion, land desertification and vegetation degradation, as well as significant changes in water resources [16–19], posing a major challenge for downstream water management. Permafrost degradation is also threatening the stability of engineering and construction in the region.

Existing studies of permafrost thermal states in the SRYR typically rely on ground temperature measurements obtained from boreholes or the active layer above permafrost, or from permafrost model simulations [7,20–22]. However, ground surface deformation is a key indicator of permafrost internal states and changes. This deformation is caused by seasonal frost heave and thaw subsidence in the permafrost environment due to the freeze-thaw cycle within the active layer. During winter, water in the active layer freezes and causes the ground surface to lift due to the difference in density between ice and water, a phenomenon known as freezing uplift. In summer, frozen water in the active layer thaws, which causes the surface to settle. These freeze-thaw cycles result in periodic ground surface deformation, and as temperatures rise, the subsurface ice in the permafrost layer begins to melt, leading to continuous ground ice melting and terrain subsidence. Both phenomena are reflected on the surface as seasonal cyclic uplift-subsidence during freeze-thaw cycles and long-term subsidence induced by ground ice melting. Therefore, recent studies have increasingly focused on monitoring surface deformation to obtain subsurface permafrost changes. Differential Interferometric Synthetic Aperture Radar (D-InSAR), a radar remote sensing technique, can obtain high-resolution and high-precision terrain deformation on a large scale. In recent years, it has been widely used for monitoring surface deformation in the permafrost environment of the QTP [23–26].

Several studies have investigated the spatial distribution characteristics of deformation in the permafrost environment in the QTP and have found that it exhibits significant spatial heterogeneity and is highly susceptible to environmental and underlying surface conditions. At the plateau scale, latitude and elevation are the primary macro factors affecting the spatial distribution pattern of permafrost and terrain deformation [27–30]. At a regional and watershed scale (10 km$^2$ to 107 km$^2$), local factors such as topography, vegetation, and soil moisture become more important in shaping the distribution pattern of permafrost [31,32]. In general, long-term subsidence is higher in warm permafrost areas with high ground ice content, indicating that warm and ice-rich permafrost is more susceptible to extensive ground ice melting. Furthermore, some studies have also detected strong subsidence/uplift signals around certain lakes, suggesting that changes in local hydrological conditions may induce localized permafrost degradation/augmentation [33–35].

In this study, we utilized Sentinel-1 SAR data and the small baseline subsets InSAR [36,37] (SBAS-InSAR) technique to obtain surface deformation time series in the SRYR. Using these time series, we extracted both seasonal deformation and long-term deformation velocity to investigate the characteristics of terrain deformation and permafrost degradation in the SRYR.

## 2. Study Area and Datasets

### 2.1. Study Area

The source region of the Yellow River is situated in the northeastern part of the Qinghai-Tibet Plateau, at the bounder of permafrost zone. It encompasses the entire area from the Yellow River's source to the Tangnaihai hydropower station's upstream, covering an area of $1.19 \times 10^5$ km$^2$, which is roughly 16% of the Yellow River basin's total area. The geographical coordinates of the study area range from 95.89°E to 103.41°E and from 32.16°N to 37.08°N. The elevation varies from 1808 to 6194 m, with an average of 3938 m, gradually decreasing from west to east. The study area has a significant number of water bodies, including large lakes, such as Gyaring and Ng¨oring Lake (sister lakes), and grouped thermokarst ponds and lakes. The total area of water bodies is 1656 km$^2$, accounting for

approximately 1.4% of the entire research area, as illustrated in Figure 1. The study area includes the western Huanghe' yan sub-basin and eastern the Aemye Ma-Chhen Range. The northern part of the Huanghe' yan sub-baisn is separated from the Buqing Mountain in the Kunlun Mountains, and the southern part is adjacent to the Bayan Har Mountains. The western boundary is adjacent to the Geshigeya Mountains [6]. The permafrost area is $5.2 \times 10^4$ km$^2$, accounting for approximately 43.8% of the total study area, while the seasonally frozen ground is $6.7 \times 10^4$ km$^2$, accounting for approximately 56.2%. The total ground ice reserves within 3–10 m depth in the SRYR is $49.62 \pm 17.95$ km$^3$, the volumetric ice content is $29.3 \pm 10.7$% on average [38].

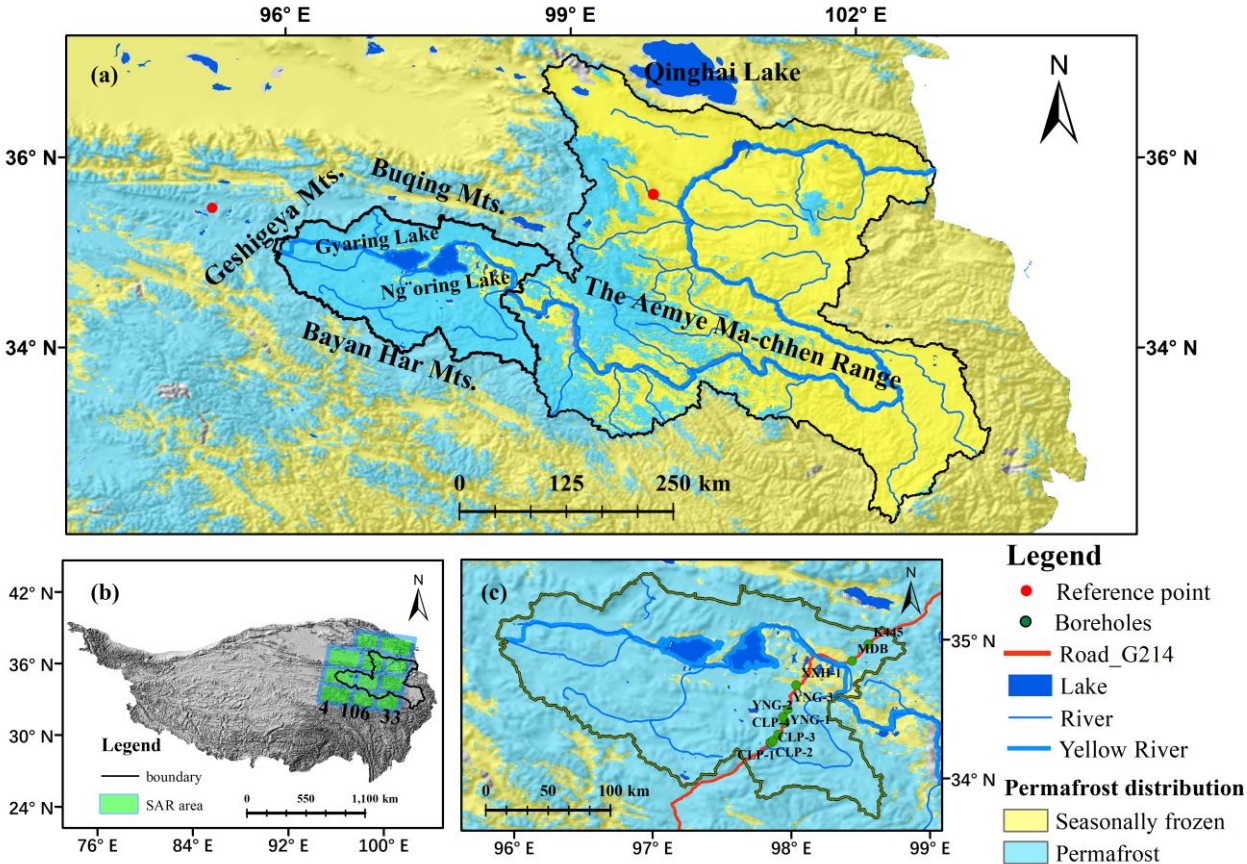

**Figure 1.** Study area. (**a**) Location of the study area on the Tibetan Plateau and coverage of SAR data (**b**) Distribution of permafrost and major mountains and water bodies in the study area (**c**) Huanghe' yan sub-basin and borehole.

The G214 national highway and the newly constructed Gonghe-Yushu highway traverse the region from northeast to southwest. The annual average temperature is −3.7 °C, and the annual precipitation is 318 mm, in range of 300–600 mm. The main vegetation types include alpine swamp meadow, alpine meadow, and alpine desert, as well as cushion vegetation and sparse vegetation on rocky beaches [7].

### 2.2. Datasets

#### 2.2.1. Sentinel-1 SAR Images

C-band Sentinel-1 SAR Level 1 single look complex (SLC) data of interferometric wide-swath (IW) mode with VV polarization were used to monitor the surface deformation. 398 Sentinel-1 SAR images, covering 3 orbits spanning from June 2017 to January 2022 were used in this study. As there is a scarcity of S1 data between 2014 and 2017 September, the S1 data before 2017 September was not adopted for InSAR analysis. Data is downloaded from Alaska Satellite Facility [39] (https://vertex.daac.asf.alaska.edu, accessed on 1 April 2022).

The three orbits cover the extent of permafrost-affected region in the source region of the Yellow River, with detailed coverage of three orbits displayed in Figure 1b and information listed Table 1.

**Table 1.** Information on Sentinel-1 images used in this study for deriving ground surface deformation.

| Path Number | Frame Number | Acquisition Period | Path |
|---|---|---|---|
| 4 | 467–481 | 2014.10.29–2022.01.08 | Descending |
| 33 | 467–485 | 2014.10.07–2021.12.29 | Descending |
| 106 | 466–482 | 2014.12.11–2021.12.16 | Descending |

To study the deformation of permafrost in the SRYR, this paper utilized 398 Sentinel-1 SAR images, mainly covering the period between June 2017 and January 2022. Of these images, 122 were from orbit 4th, 145 from orbit 106th, and 131 from orbit 33rd, resulting in a total of 1185 pairs of interferometric SAR images generated from the experimental data.

2.2.2. Underlying Surface

As shown in Figure 2, this study examined the impact of four types of underlying surfaces on deformation in the permafrost boundary region of the northeastern Tibetan Plateau. The four types of underlying surfaces studied were: Mean Annual Ground Temperature (MAGT) [13], ground ice content [9], Normalized Difference Vegetation Index (NDVI), and Topographic Position Index (TPI). This region is mainly comprised of high-temperature permafrost, which makes it highly susceptible to warming climates [6,40]. MAGT was selected as a factor for analyzing the deformation characteristics of permafrost with different thermal states, as it reflects the permafrost's thermal state. Vegetation significantly influences heat exchange between air and ground surface, thereby exerting a strong effect on permafrost thermal state. As a result, NDVI was also utilized in this study as a quantitative indicator for vegetation development. TPI was used to describe the topographic position and was used as a terrain factor for analyzing the deformation characteristics of different slopes.

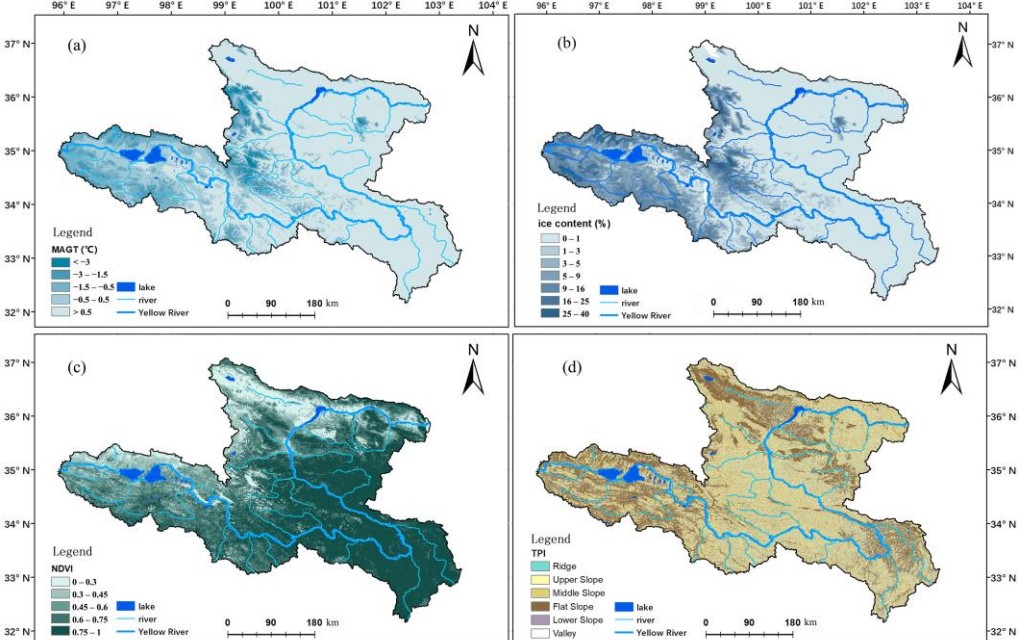

**Figure 2.** Underlying surface in the SRYR (**a**) Mean Annual Ground Temperature (MAGT) (**b**) volumetric ground ice content (**c**) Normalized Difference Vegetation Index, NDVI (**d**) Topographic Position Index (TPI).

MAGT [3] data used in this study was derived mainly from a modified MODIS surface temperature (LSTs) freezing and thawing index from 2005–2015. Combined with the Temperature at the Top of Permafrost (TTOP) model, it was validated using various surface datasets. Ground ice content [9,41] refers to the permafrost ice reserves on the Qinghai-Tibet Plateau, which was estimated based on the Quaternary sediment type map and the newly developed permafrost distribution and permafrost thickness maps, utilizing data from 164 boreholes from 2003–2012. NDVI data was calculated from Sentinel-2 (S2) visible and near-infrared bands (B4 and B8) using Google Earth Engine, with the average value taken during 2017 and 2022 at a spatial resolution of 10 m. TPI is a terrain parameter used to describe the topographic position. Its fundamental concept is to determine the position of a point on a slope based on the difference between its elevation and the average elevation within a certain range around it, combined with the slope at that point.

### 2.2.3. SRTM DEM

In this study, the Shuttle Radar Topographic Mission (SRTM) digital elevation model (DEM) (http://gdex.cr.usgs.gov/gdex, accessed on 1 April 2022) was utilized to calculate the slope as well as remove the topographic phase and implement geocoding during InSAR processing.

### 2.2.4. Borehole Data

The borehole data used in this study were obtained from previous research conducted by Luo et al. [20]. The boreholes selected were located on both sides of the Bayan Har Mountains along the G214 Highway and were distributed in a north-south direction. Ten boreholes used in this study are identified and labeled in Figure 1c, and their information is presented in Table 2.

**Table 2.** Information of boreholes along the National Highway G214 in the Headwater Area of the Yellow River during 2010–2016. Boreholes are listed from south to north, and the temperature status were from [20].

| Borehole Name | Longitude (°E) | Latitude (°N) | Elevation (m) | Rate of Change (°C a$^{-1}$) | Surface Vegetation (Alpine-) | Average MAGT 20 m (°C) | Velocity (mm/a) | Seasonal Deformation (mm) |
|---|---|---|---|---|---|---|---|---|
| CLP-1 | 97.8 | 34.2 | 4721 | 0.014 | swamp meadow-dense vegetation | −1.67 | −20.3 | 25.2 |
| CLP-2 | 97.8 | 34.2 | 4724 | 0.003 | swamp meadow-dense vegetation | −1.57 | −17.3 | 25.7 |
| CLP-3 | 97.9 | 34.2 | 4663 | 0.009 | meadow-dense vegetation | −1.00 | −12.4 | 10.4 |
| CLP-4 | 97.9 | 34.3 | 4564 | 0.019 | swamp meadow-dense vegetation | −0.50 | −12.3 | 9.9 |
| YNG-1 | 97.9 | 34.4 | 4446 | 0.001 | meadow-moderate vegetation | 0.03 | 2.6 | 6.9 |
| YNG-2 | 97.9 | 34.4 | 4395 | 0.017 | steppe-sparse vegetation | 1.25 | 2.6 | 7.4 |
| YNG-3 | 97.9 | 34.4 | 4324 | 0.028 | steppe-sparse vegetation | 1.15 | 11.8 | 14.3 |
| XXH-1 | 98.0 | 34.6 | 4221 | 0.190 | swamp meadow-dense vegetation | 0.87 | 0.1 | 17.7 |
| MDB | 98.4 | 34.8 | 4225 | 0.007 | steppe-sparse vegetation | −0.58 | −15.9 | 5.2 |
| K445 | 98.5 | 34.9 | 4282 | 0.014 | steppe-sparse vegetation | −0.89 | −37.3 | 11.9 |

## 3. Methods

### 3.1. SBAS-InSAR Processing

3.1.1. Generation of Coregistered Stack of SLC Images and Interferograms

For each orbit, one of the images is selected as the master image, and the other satellite data are aligned to this image to generate a multi-looking differential interferogram. Generate differential interferograms for master and slave SAR images at different times. All the SLC images were coregistered to the stack reference. After generating a coregistered stack of SLC images, interferograms were generated by each SAR image with its two sequential acquisitions [37,42], i.e., for Scene n, with Scene n + 1 and n + 2. We performed multi-looking with 25 pixels in range and 6 pixels in azimuth to form a square pixel (~90 m in ground resolution) and to reduce the noise. The differential phase was unwrapped using the minimum cost flow algorithm (SNAPHU) [43]. Figure 3 shows the network of unwrapped differential interferogram pairs were converted to the displacement time series by applying a weighted least squares estimator. In the process of estimating the time series deformation, the inverted interferogram network after unwrapping is used to construct the timeline of the line-of-sight (LOS) displacement map. The interferogram network is reversed into a time series by applying a weighted least squares (WLS) estimator [44], where the interferogram is weighted by the inverse of the phase variance. Unlike some studies that presuppose a deformation model to facilitate solving the phase time series in permafrost environments, we did not preset any deformation in this study. The original phase time series were solved by minimizing the phase residuals in the WLS estimator. The first scene image in the dataset was used as the reference for the LOS displacement time series. The measurement of the absolute distance due to surface deformation is not accurate.

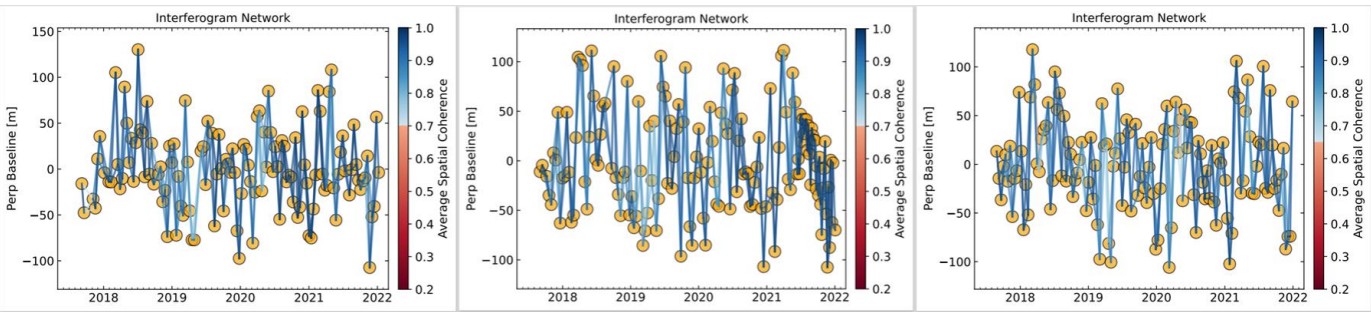

**Figure 3.** Interference network diagram for each orbit, from left to right, 4th, 106th, 33rd orbits respectively.

3.1.2. Georeferencing Processes

All measurements are performed for the reference point. The information on the deformation of the surface with respect to the reference point is obtained using InSAR. We usually choose areas with high coherence (e.g., exposed bedrock) as reference points. Pixels with extremely high temporal coherence were selected as reference points (as shown in Figure 1) to compare the spatial differences in surface deformation. Then, correct tropospheric atmospheric phase delay due to different tropospheric atmospheric conditions in two SAR images using atmospheric model simulated interferometric images [45]. Finally, methods such as ground delay correction, phase attenuation, and terrain residual correction were used to reduce residual surface deformation [46]. The deformation time series were geocoded into the WGS84 coordinate system with a resolution of 0.001 × 0.001 degrees and reprojected into the Albers Equal Area Conic system of 100 m grid spacing, and central meridian was 95°. Two reference points are used to adjust the spatial deformation information. One is on the bedrock on the north side of the mountain top on overlapping region of orbit 106th and 33rd. This area has a dry and homogeneous surface with high coherence close to 1. The second reference point is on the northwest area of the Geshigeya mountain,

which is on the overlapping area of orbit 106th and 4th. After acquiring deformation time series, a moving window filter with a size of 3 was applied to the deformation time series pixel by pixel reduce the impact of outliers.

### 3.1.3. Impact of the Earthquake on Deformation

Worth of attention is that 7.4-amplitude earthquake occurred in Maduo County, Guoluo Tibetan Autonomous Prefecture, Qinghai Province, China at 02:04 on 22 May 2021 [47], which had a significant impact on the deformation time series results in the scenes around this time period. The monitored deformation in the vicinity of the earthquake and near this seismic event is significantly influenced by the seismic activity. Since we are interested in the deformation caused by permafrost freeze-thaw activities, we extracted and eliminated the deformation segments within this period (i.e., two SAR data acquisition intervals following the earthquake) from the time series. We assumed that any deformation caused by permafrost during this time is small and can be neglected and concatenated the subsequent deformation time series after the earthquake to the end of the deformation time series preceding the earthquake (details seen Figure A2).

### 3.2. Long-Term Deformation Velocity and Seasonal Deformation Calculation

After obtaining the time series deformation, the deformation process signal contains the interannual deformation rate and seasonal deformation amplitude. To derive the long-term deformation trend, a linear trend is usually used to calculate the long-term trend rate. For seasonal deformation, the maximum-minimum geopotential difference is calculated for each year and the values are then averaged over many years to represent the average state over the survey period. The analysis involves a sinusoidal seasonal function, together with a linear trend in the time series of each image pixel displacement, which can be expressed mathematically as:

$$D(t) = v \cdot t + A \cdot \sin\left(\frac{2\pi}{T} \cdot t + \varphi\right) + C \tag{1}$$

where $A$ is the periodic seasonal amplitude, $T$ is the period of seasonal fluctuations (assumed to be one year), $\varphi$ is the initial phase, and c is the residual phase. $v$, $A$, $\varphi$, and $C$ are determined from the deformation time series and solved in a spatial grid on a pixel-by-pixel calculation. The periodic seasonal deformation is twice the amplitude $A$.

Nash-Sutcliffe efficiency coefficient (*NSE*) is commonly used to quantify the prediction accuracy of simulation models (e.g., hydrological models). It indicates the fitness of the model to the deformation time series whether the model can reproduce the SBAS-InSAR-monitored deformation time series. The model *NSE* values of the three orbits are presented in Figure A1.

$$NSE = 1 - \frac{\sum\left(y_i - y_i^{pre}\right)^2}{\sum(y_i - \bar{y})^2} \tag{2}$$

where: $y_i$ is the deformation at date i; $\bar{y}$ is the mean value of deformation. $y_i^{pre}$ is the model fitted deformation at date *i*. *NSE* approaching 1 indicates good accuracy of results.

## 4. Result

### 4.1. Deformation Characteristics along the Profile

The study area exhibits a notable variation in elevation between its western and eastern parts, with higher elevations found in the west and lower elevations in the east. This elevation difference is most clearly visible in the north-south direction of the G214 national highway as it crosses the Yellow River.

The deformation characteristics of the valley where the Yellow River flows through, as well as its northern and southern mountainous regions, differ from each other. According to the permafrost distribution map, both the northern and southern regions belong to the permafrost zone, while the middle area where the Yellow River flows through is in the

seasonal frozen zone. Comparing the deformation characteristics between point B and point C, it was found that the annual deformation rates between the two points were not significantly different, with both being around 30 mm/a, but the seasonal deformation amplitude was smaller at point B and larger at point C. As these two points lie on the north and south sides of the Yellow River with a large difference in elevation between them, a profile line was established between points B and C to explore the influence of elevation on the deformation differences between the two points. Figure 4c shows the relationship between long-term deformation velocity and elevation along the profile line between points B and C. The left vertical axis represents elevation (m), while the right axis represents the long-term deformation velocity (mm/a). The permafrost zones on the north and south sides are marked in the figure. The results showed that there was a certain negative correlation between long-term deformation velocity and elevation. The deformation rate in the middle region displayed an uplift trend with smaller absolute values, while the deformation rates in the northern and southern regions were higher than those in the middle area, showing a trend towards subsidence. Despite the differences in elevation between the two areas, points B and C exhibited similar characteristics in terms of long-term deformation velocity. The subsidence trend on the profile line increases from north to south, then decreases, and becomes uplifted in the valley area before turning to a southward subsidence. The analysis suggests that the increase in subsidence trend from point B to the south is due to the influence of slope position factors, as the southern region of point B is a gentle area.

Figure 4b depicts the relationship between seasonal deformation and elevation along the profile line between points B and C. The left vertical axis indicates elevation (m), while the right axis represents the seasonal deformation amplitude (mm). The amplitude on both sides of the valley is higher than that in the middle, but the degree of change is not uniform. The seasonal deformation amplitude near point C is significantly greater than that near point B on the north side. The NDVI value near point C is 0.73, while the NDVI value near point B is 0.39, indicating that vegetation affects the water-holding capacity of the surface. This results in the water content of the active layer changing with the seasons, thereby affecting the seasonal variation of the surface in the region. Figure 4a shows that there are numerous lakes and ponds visible on the surface from the Google Earth image. The terrain here is relatively flat, with abundant surface water and ground ice.

### 4.2. Long-Term Deformation Velocity

Figure A1 in the study shows the fitness of the deformation model to the InSAR-monitored deformation time series. Generally, areas with high values of deformation, such as the Huanghe'yan sub-basin, have higher NSE values.

Furthermore, the spatial distribution of long-term deformation velocity in the study area is presented in Figure 5. The results were obtained by combining the outcomes of three tracks in the LOS direction. The areas in red represent subsidence, while those in blue indicate uplift. The results from tracks 4th and 106th are consistent in the overlapping areas, and no discontinuities or jumps were detected in the splicing area. This confirms the robustness of the velocity signals extracted from the two tracks, despite their different acquisition dates. However, some unclear splice traces were observed at the junction of tracks 106th and 33rd, due to the two tracks having different reference points (as shown in Figure 1a) and the influence of the earthquake.

The long-term surface deformation trends in the study area range from $-65.8$ mm/a to 74.6 mm/a, with an average annual deformation rate of $-4.2$ mm/a. Approximately 29.8% of the study area has relatively stable ground surface, with deformation velocities of $-2$ to 2 mm/a. It is worth noting that areas with slopes greater than 10 degrees were excluded during the statistics to prevent the impact of large slopes on the accuracy of the surface deformation retrievals. Uplift signals were also observed in the study area, mainly concentrated near the Yellow River channel on the eastern side of Lake Gyaring and between the northern and southern mountains. These uplift signals may be due to the footslope deposition and tectonic movement, such as that caused by the Maduo earthquake.

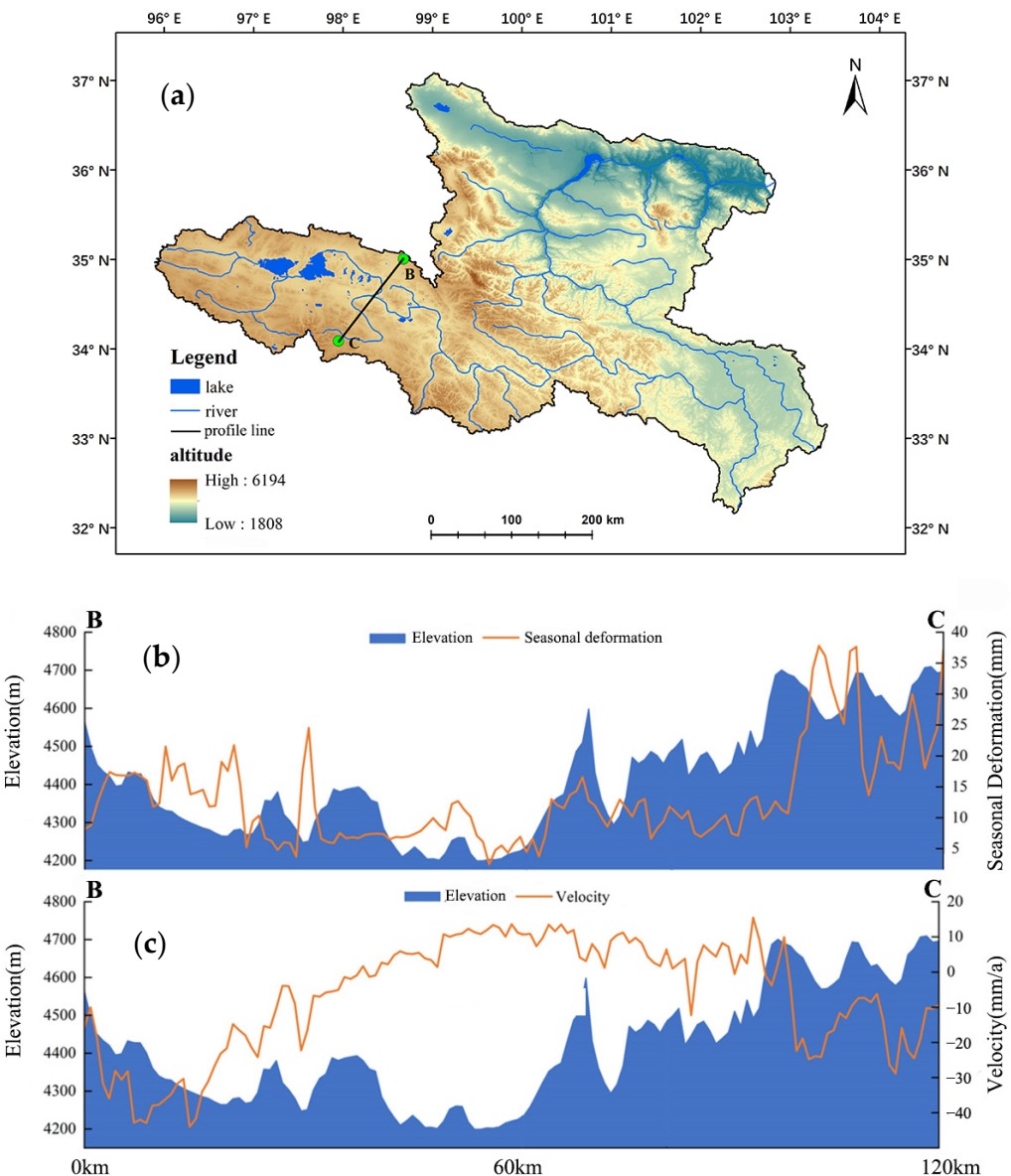

**Figure 4.** (**a**) Elevation of the entire study area (**b**) Profile line between seasonal deformation and elevation between points B and C (**c**) Profile line between long-term deformation velocity deformation and elevation between points B and C.

### 4.3. Seasonal Deformation

Figure 6 illustrates the spatial distribution of seasonal deformation in the SRYR in the LOS direction, which was stitched from three tracks. The seasonal deformation can reach 40 mm, with an average of 8.85 mm. The seasonal surface deformation in permafrost regions is significantly higher. Areas with seasonal deformation exceeding 50 mm are particularly large in the western Bayan Har Mountain north of Gyaring Lake and the boundary of the study area in the south of Ng¨oring Lake. The analysis also revealed that the area in Huanghe'yan sub-basin near Bayan Har Pass exhibits significant subsidence, with an annual deformation rate of up to −40 mm/a. Meanwhile, the Bayan Har Mountain north of Gyaring Lake and the Chalaping area of the Bayan Har Mountain experience significant cyclic freeze uplift and thaw subsidence, with seasonal deformation exceeding 50 mm.

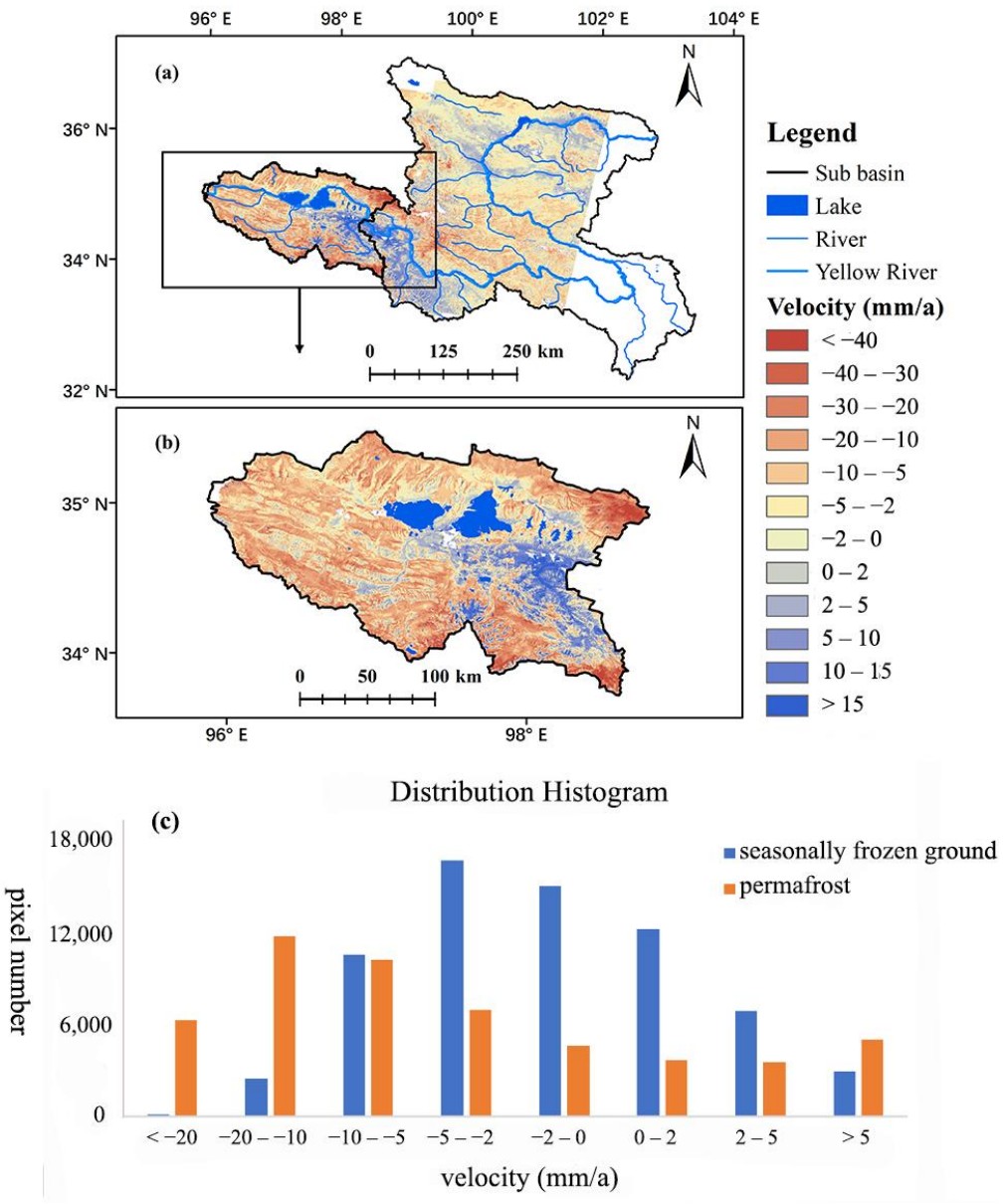

**Figure 5.** (**a**) Spatial distribution of long-term deformation velocity of study area (**b**) Spatial distribution of long-term deformation velocity of sub basin. Negative values indicate that the surface moves away from the satellite direction, which indicates subsidence, while positive values indicate movement towards the satellite direction, indicating uplift. Lake and glacier areas are masked out from the deformation map (**c**) Distribution histogram of long-term deformation velocity for the entire SRYR.

### 4.4. Deformation Distribtion Characteristics in Different Underlying Surface

Figures 7 and 8 present the distribution pattern of long-term deformation velocity and seasonal deformation in different subsurface environments. It is evident that the deformation on flat slopes is significantly greater than that on other slopes (as shown in Figures 7d and 8d). Flat slopes cover about 29.5% of the area, with an average seasonal deformation amplitude of 12.7 mm and an average annual deformation rate of −9.9 mm/a, much higher than the average deformation magnitude of the entire SRYR.

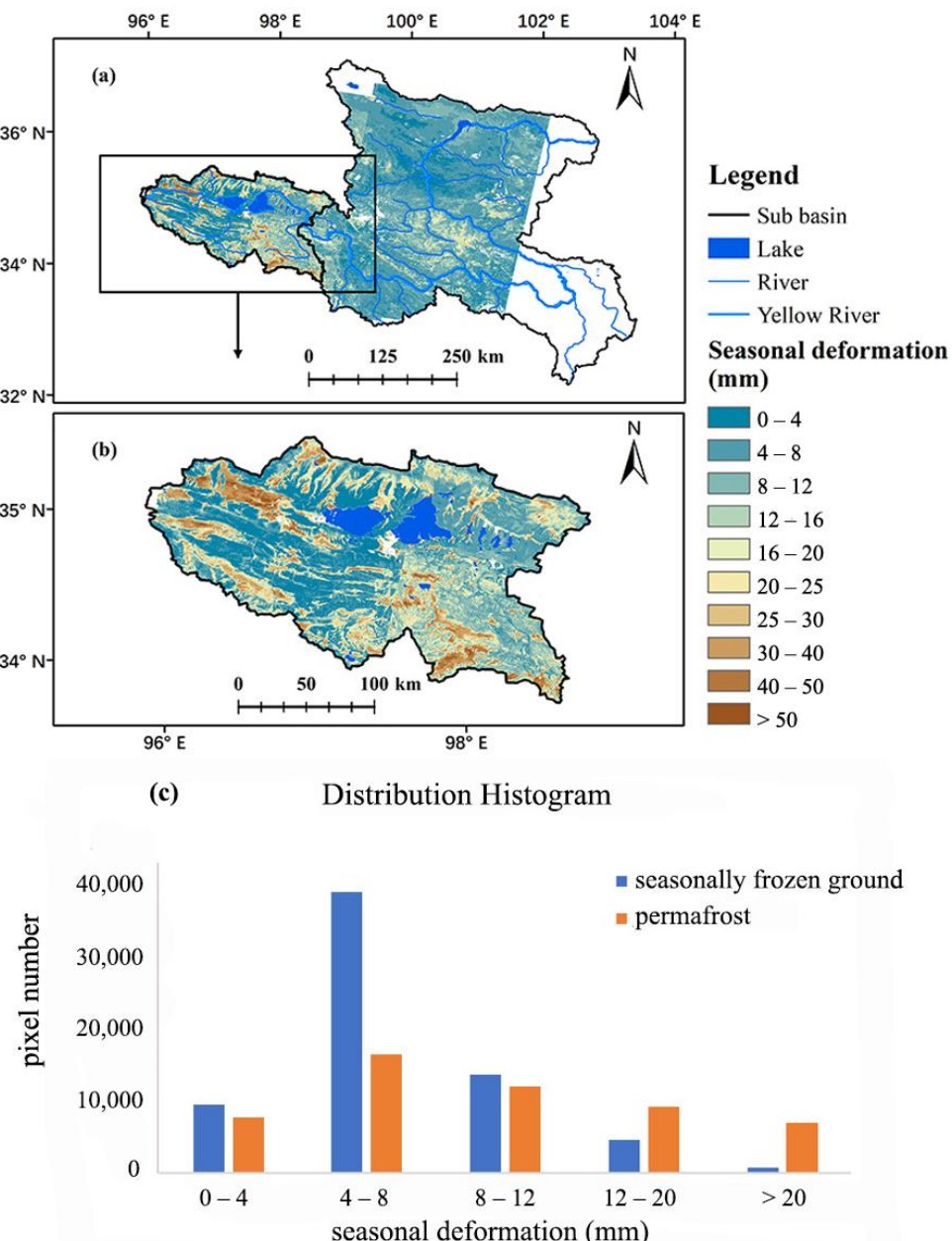

**Figure 6.** (**a**) Spatial distribution of seasonal deformation of study area (**b**) Spatial distribution of seasonal deformation of sub basin (**c**) Distribution histogram of seasonal deformation for the entire SRYR.

Figure 7b illustrates that there is a positive correlation between the absolute value of annual deformation rate and ground ice content (Pearson correlation coefficient 0.28 and *p*-value < 0.005), with an increasing subsidence trend as the ground ice content increases.

In the study area, permafrost is in a degrading state, and the subsidence trends intensify with the increase of MAGT. There is a certain positive correlation between the absolute value of the annual deformation rate and MAGT below −0.5 °C (as shown in Figure 7a). In areas with MAGT < −0.5 °C, the correlation coefficient between MAGT and long-term velocity reached −0.18 (*p*-value < 0.005). The subsidence velocity tends to be the largest in the region with an annual average ground temperature of −1 °C∼−0.5 °C with an average annual deformation rate of −10.1 mm/a, a median of −9.8, and a standard deviation of 10.3.

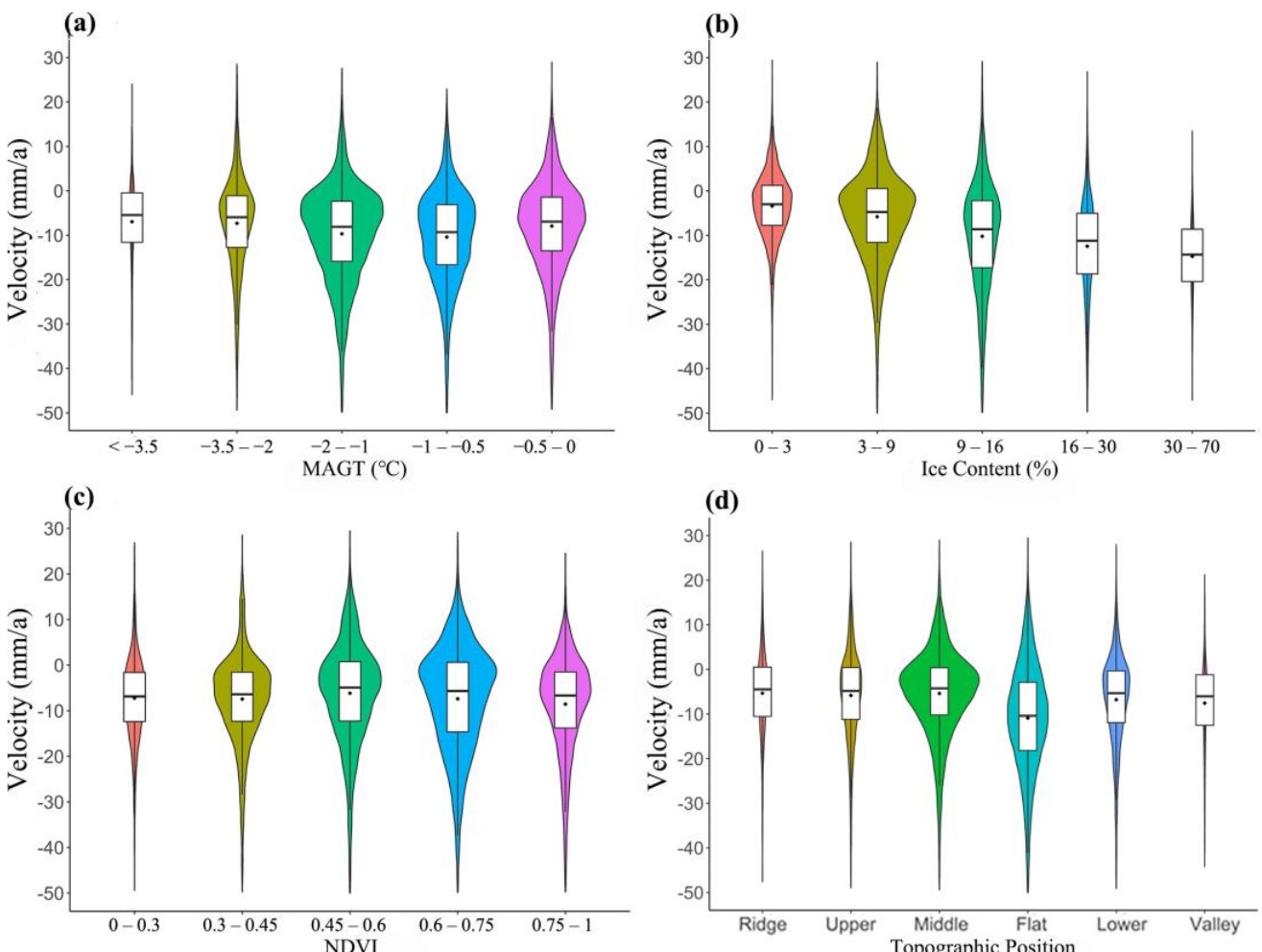

**Figure 7.** Long-term deformation velocity results for different underlying surface.

The relationship between the seasonal deformation and MAGT is positive, with the seasonal deformation increasing as MAGT increases (as shown in Figure 8a). This trend is less pronounced in areas with temperatures below $-2\,°C$, while significant increases in seasonal deformation are observed when the temperature is higher than $-2\,°C$. In areas with an annual mean ground temperature between $-1\,°C$ and $-0.5\,°C$, the average seasonal deformation amplitude is 11.0 mm, with a median of 9.0 and a standard deviation of 9.2.

The study also found a positive correlation (Pearson correlation coefficient of 0.2 and $p$-value $< 0.005$) between the seasonal deformation and ground ice content, with the seasonal deformation increasing as the ice content increases. Areas with higher ground ice content typically have sufficient water supply in the active layer, resulting in significant seasonal deformation.

The relationship between the NDVI and absolute value of annual deformation rate is not linear, showing a trend of first decreasing and then increasing (as shown in Figure 7c). The impact of vegetation on the long-term subsidence trend of the permafrost terrain is mainly through its influence on surface soil temperature and water. When vegetation conditions are good, the subsidence rate of the ground surface shows an increasing trend with an increase in NDVI value. NDVI is closely connected with active layer water content, and grounds with high vegetation coverage have more abundant water and better development of ground ice, so there is a larger possibility of high subsidence values. The study found no overall correlation between the seasonal deformation and NDVI, but did find that the seasonal deformation tends to decrease with an increase in NDVI when the NDVI value is

low, reaching a minimum in the range of 0.45–0.6, with an average seasonal deformation amplitude of 9.4 mm. The seasonal deformation then increases with an increase in NDVI, with the largest seasonal deformation amplitude observed in areas with an NDVI value greater than 0.75, with an average value of 10.2 mm.

It is important to note that the surface deformation of permafrost is the result of complex processes and is affected by many factors. Therefore, statistical correlations of each individual factor are not always significant. Nonetheless, Figures 7 and 8 illustrate clear trends in the deformation patterns of different underlying surfaces.

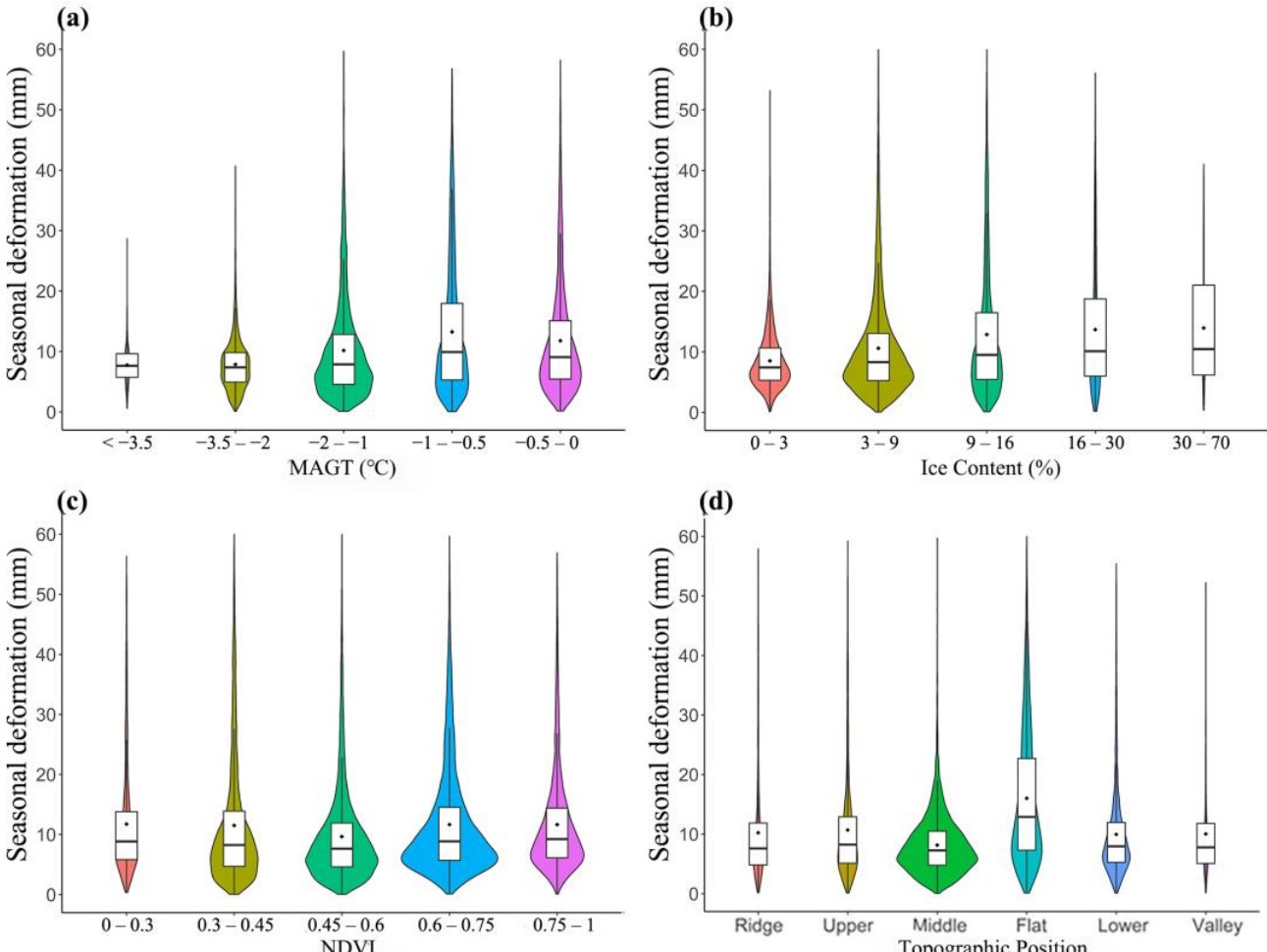

**Figure 8.** Seasonal deformation results for different underlying surface.

## 5. Discussion

### 5.1. Relationship between Deformation and Permafrost Status at Boreholes

The Huanghe'yan sub-basin contains the highest permafrost coverage in the study area and shows large spatial heterogeneity in deformation. Figure 9 displays the time series deformation curves for eight out of ten drilling boreholes in this sub-basin. The borehole locations are marked in Figure 1c, and their temperature status, vegetation coverage, and deformation are summarized in Table 2. These boreholes provide insights into the factors contributing to the observed deformation patterns, highlighting the complex interplay between various environmental variables such as ground temperature, vegetation, and surface terrain condition.

Among the borehole sites, CLP1, CLP2, CLP3, CLP4, K445, and MDB, located within the permafrost zone, exhibit clear subsidence. The CLP four points, located in the southern Bayan Har Mountains Chalaping area, exhibit very clear periodic frost heave and thaw subsidence patterns and long-term subsidence trends. CLP1 point has a long-term

subsidence velocity reaching 20.3 mm/a and seasonal deformation magnitude reaching 25.2 mm. The subsidence trend and the seasonal deformation magnitude both gradually decrease from south to north. The northern K445 point and the MDB point in the profile show significant long-term subsidence trends but less obvious seasonal deformation, with a long-term deformation velocity of −15.9 mm/a for MDB point and −37.7 mm/a for K445 point.

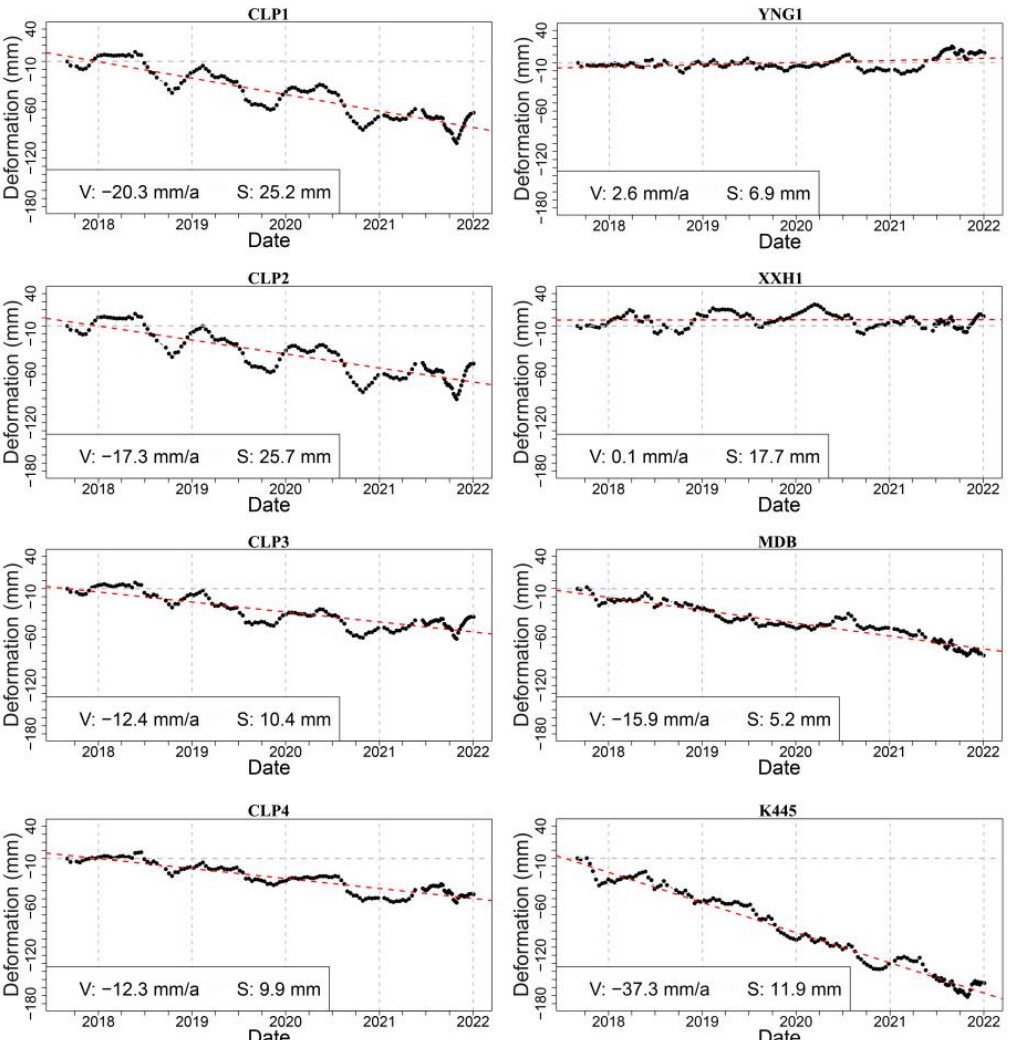

**Figure 9.** Time series deformation curves of boreholes. V represents for velocity and S represents for seasonal deformation.

The greater seasonal deformation magnitude at CLP1 and CLP2 compared to MDB and K445 is attributed to dense surface vegetation, particularly in the alpine swamp meadow with abundant surface water in the southern Chalaping area. The study area's vegetation is well-developed, with nearly half of the area having an NDVI above 0.6. The surface vegetation is usually alpine swamp meadow, which indicates sufficient soil moisture. The high soil moisture content in the active layer in this region results in a large frost heave and thaw subsidence seasonal deformation. The amplitude is also high when the NDVI value is low because the surface lacks vegetation cover, usually bare or alpine desert, resulting in direct heat transfer to the active layer, causing a thicker active layer. The subsidence rate at CLP1 and K445 is much higher than other points, which can be linked to the significantly larger MAGT increasing rate in these locations. This indicates a trend of increasing MAGT and significant permafrost degradation, which is reflected in the deformation signal as a higher interannual subsidence rate.

In the middle of the profile, in the Yellow River valley, YNG three points and XXH1 exhibit an uplift trend. Points YNG1 and YNG2 show an uplifting trend of 2.9 mm/a, and point XXH1 exhibits a negligible trend of 0.1 mm/a. YNG-3 in the middle of the valley shows a large uplift trend, which may be due to the deposition of sand and gravel on both sides of the slope or variant soil water along the Yellow River valley.

### 5.2. Permafrost Boundary Refinement Using Deformation Information

The eastern region of the Yellow River source area is located at the boundary of permafrost on the Qinghai-Tibet Plateau. However, the accurate delineation of this boundary is not clear yet. The following Figure 10 illustrates three widely used distribution maps of permafrost in the study area. The delineation of this boundary is not consistent among the three maps [3,13,14]. Figure 10a successfully distinguishes permafrost and seasonal frost at the profile location, but it is evident that the permafrost extents in Figure 10b,c are much wider and extend further east. Figure 10b has the largest permafrost extent but seems overestimates its actual range. Our deformation results also indicate that the large and continuous deformation occurs in the continuous permafrost zone in Huanghe' yan subbasin, but there are scattered deformation signals in the eastern part of the basin. In this boundary region, permafrost was formed under a favorable climate, in some areas, under warmer climatic condition, permafrost can still survive as it protected by the ecosystem environment. For example, Figure 10d shows the image of the black rectangular area in Figure 10a–c on Google Earth, which illustrate a small wetland/peatland region where two permafrost maps indicated no presence of permafrost, showed a high magnitude of both seasonal deformation and long-term subsidence, which indicates the existence and degradation of permafrost. The frozen to unfrozen thermal conductivity ratio of wet organic soil is about eight times greater than that of dry gravel, and wet silt is about four times greater [48]. The ecosystem provides favorable conditions for the development and preservation of permafrost in this region. In further studies, in permafrost boundary regions, the integration of deformation information, temperature, and vegetation data would serve to better refine the delineation of regional permafrost boundaries.

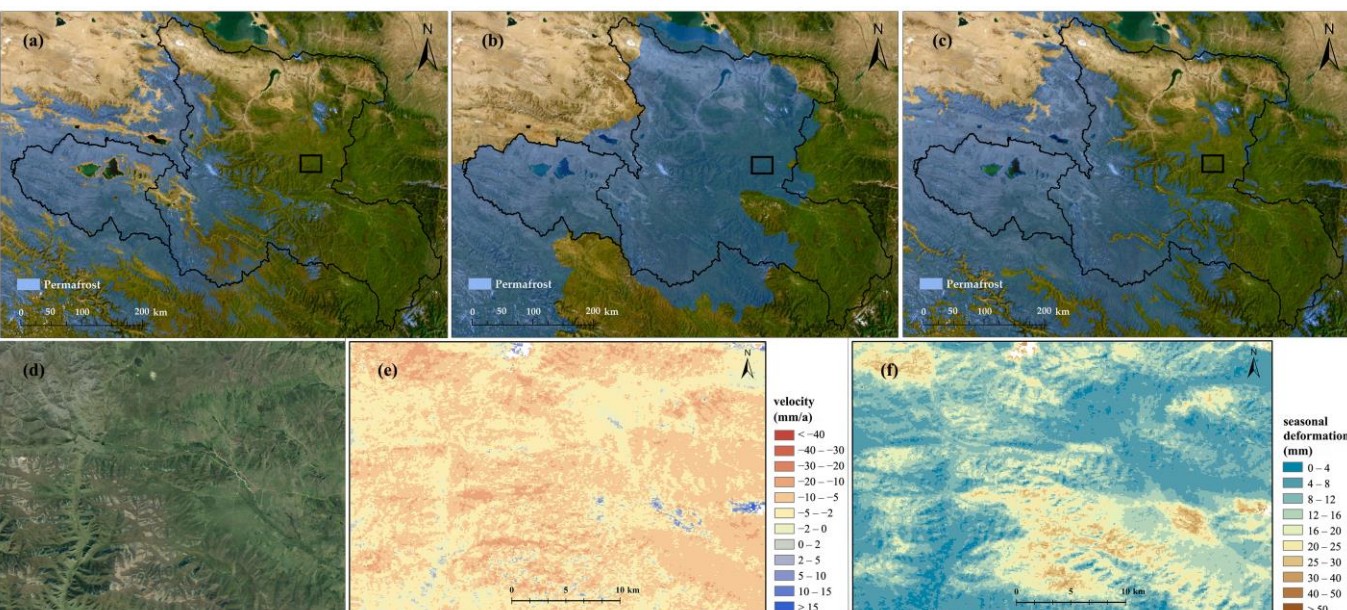

**Figure 10.** Three widely-used distribution maps of permafrost in the study area: (**a**) Zou2017 [3] (**b**) Ran2019 [13] (**c**) Obu2019 [14] (**d**) Google Earth image of the black rectangular area (**e**) Long-term deformation of the black rectangular area (**f**) Seasonal deformation of the black rectangular area.

## 6. Conclusions

This study investigates the deformation characteristics in the Source region of the Yellow River in the northeast part of the Qinghai-Tibet Plateau, utilizing Sentinel-1 satellite SAR data from 2017 to 2022 and SBAS-InSAR technique. The findings are as follows:

The overall trend in the Source region of the Yellow River is subsidence, with an average long-term deformation velocity of −4.2 mm/a, and the average long-term deformation velocity in the permafrost area is −7.3 mm/a, which is significantly higher than the average level of the entire watershed. The high-value areas of long-term deformation velocity are in the Huanghe' yan sub-basin, which has the highest permafrost coverage. The Huanghe' yan sub-basin also manifests high spatial heterogeneity in deformation. The northern and southern sides of the Yellow River, the northern side of the Bayan Har Mountains, and the eastern part of the Buqing Mountains show obvious subsidence trends. The average seasonal deformation magnitude of the Source region of the Yellow River is 8.85 mm, with an average seasonal deformation of 11.2 mm in the permafrost area, and the high values are concentrated in the northern side of the Bayan Har Mountains and the headwaters of the Yellow River in the northwest of the Gyaring Lake.

In SRYR, environmental factors, such as ground temperature, ground ice content, vegetation type and topography all affect the deformation. Topography strongly affects ground surface deformation, with flat slopes has much higher subsidence rates and seasonal deformation. Ground temperature and ground ice richness also exhibit a certain degree of influence on the deformation pattern. And there is no overall correlation between deformation and NDVI.

At Huanghe' yan sub-basin, which has a high permafrost coverage and high deformation heterogeneity, we analyzed the deformation details at eight boreholes and a profile line covering different underlying surface conditions. Permafrost boreholes all show subsidence trends, while the seasonally frozen ground in the river valley manifests a small uplift trend. Areas with significant deformation characteristics have a high similarity to the distribution range of permafrost, indicating that deformation has great potential in reflecting permafrost distribution. This study also found that certain wetland/peatland regions might have the potential of permafrost existence, as indicated by deformation characteristics. Terrain deformation could serve as a data support for permafrost distribution mapping in the permafrost boundary region in future studies.

**Author Contributions:** Conceptualization, C.L. and L.Z.; methodology, C.L. and L.W.; formal analysis, C.L. and L.W.; resources, L.Z., E.D., D.Z. and G.L.; data curation, C.L., L.W., H.Z. and S.L.; writing-original draft preparation, C.L.; writing-review and editing, C.L., L.W., L.Z., Z.L. and Y.H.; project administration, L.Z.; funding acquisition, L.Z. and L.W. All authors have read and agreed to the published version of the manuscript.

**Funding:** This research was supported by the Second Tibetan Plateau Scientific Expedition and Research (STEP) program (No. 2019QZKK0201), the Natural Science Foundation of Jiangsu Province (BK20200828), and research grants from the National Natural Science Foundation of China (No. 41931180 and 42001054).

**Data Availability Statement:** The Sentinel-1 SAR data was downloaded from the Alaska Satellite Facility (https://www.asf.alaska.edu/sentinel/, accessed on 1 April 2022). The TPI data source was the Chinese Academy of Sciences Computer Network Information Center's Geographic Spatial Data Cloud Platform (http://www.gscloud.cn, accessed on 1 April 2022).

**Conflicts of Interest:** The authors declare no conflict of interest.

## Appendix A

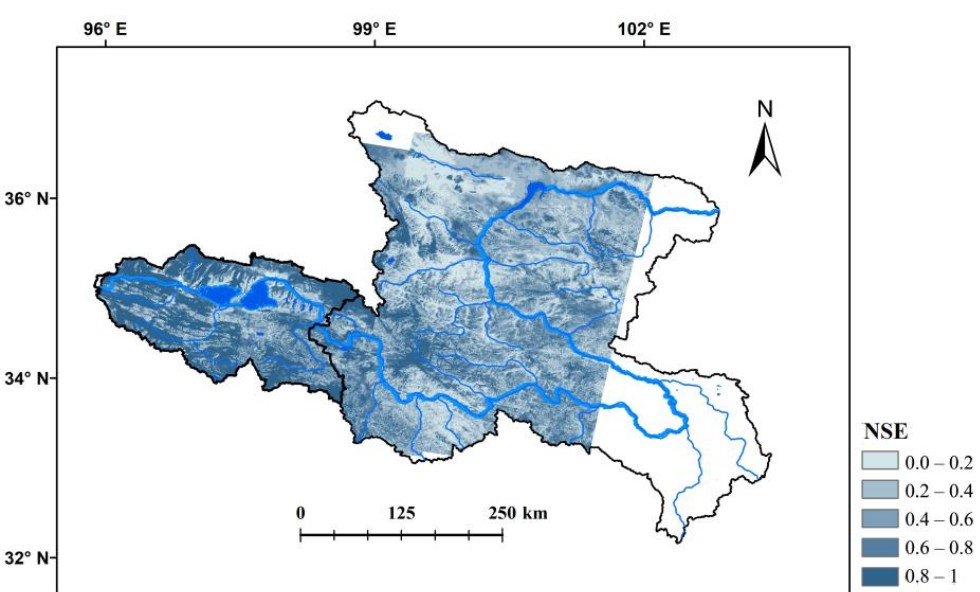

**Figure A1.** Model NSE values, which indicates the fitness of the deformation model to the InSAR-monitored deformation time series.

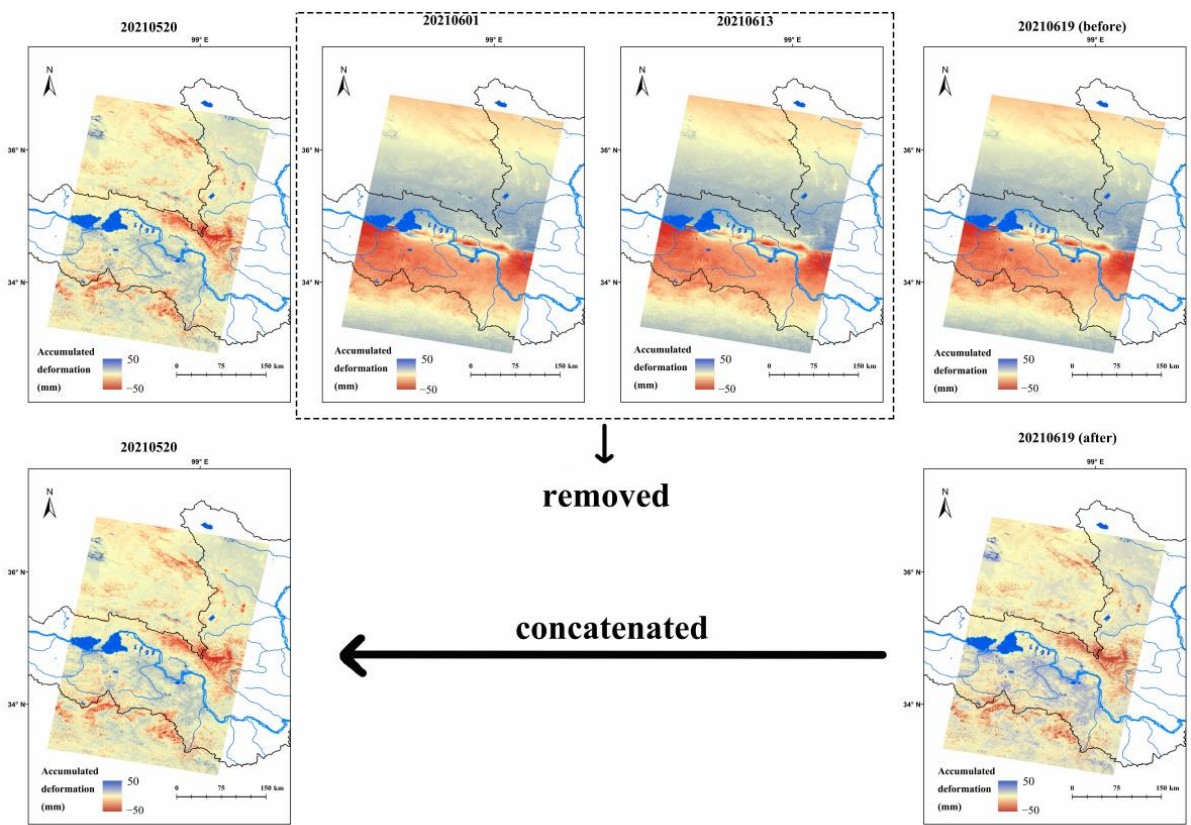

**Figure A2.** Illustration of processing scheme for 106th orbit affected by earthquake. The upper panel shows the original deformation results of four dates near of earthquake occurrence; the bottom panel shows the post processed deformation results. It assumed that the deformation during 20210520 (the scene before earthquake) and 20210601(the scene after earthquake) is small and can be neglected. We extracted the deformation during 20210619 and 20210613, and then concatenated the subsequent deformation time series after 20210619 to the end of the 20210520.

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
