# Peer review of "Ground Deformation and Permafrost Degradation in the Source Region of the Yellow River, in the Northeast of the Qinghai-Tibet Plateau"

_remotesensing, doi:10.3390/rs15123153_

Round 1

Reviewer 2 Report

Dear Authors,

Here are my comments on your manuscript. It needs major revisions before a final assessment and decision can be made.

Please focus on better defining the specific scale of analysis, which is irrevocably bound to uncertainty and accuracy estimations. This is the confidence core of what you offer to Readers.

--

Comments per line number:

1. Terrain includes what? Are you talking about surface+subsurface? Why not using just subsidence/thermokarst if that's what you measured?

1 and 6. You either avoid acronyms for source region Yellow River--- in Abstract or put them accordingly

19-- SRYR?

37. What is large?

38. What is SAYR?

42-43. According to whom? Put reference.

43-51. Has active-layer thickening cooled down surface flows as now they run deeper? I would not ask this if you hadn't put in line 48 'ecological degradation' which is a broad concept.

53. Ground temperature? 0.6m ?

56. What is the season here? Snow-season length important?

59. If you are talking about ground thaw, don't use melt.

60-61. Keep it simple, just say active-layer thaw...

59-65. You could just say freeze-thaw cycles cause uplift and subsidence which is quite obvious but try to rephrase it more clearly so that you can establish that it will be your proxy for permafrost changes. Nevertheless, you need to always have the certainty that you are talking about a 100% permafrost profile, and not a seasonal-frost profile.

68. What is the vertical resolution?

71. How many references you need to state this? Keep only necessary references.

72. Again, it would be great to have a simple scale-related definition for deformation since, cryoturbation processes could be understood as a deformation of ground-structure, but do not represent the type of deformation you're measuring in your study.

72-84. Define scales since you're methods are bound to the implicit hypotheses on deformation you have proposed. Approximate ranges of slope and vertical changes.

85. What is small?

108. correct to '3-10 m'

108-109 Bad citation here: Ref. [36] 10.1109/JSTARS.2017.2671025 does not refer to SRYR. Please re-check all references.

112. Punctuation missing.

115. Put regional map first, then the regional zoom ins.

115. Briefly explain in Figure1's caption what are 'reference points' for.

Please consider that a minimal geodetic control density of 0.001:1 (0.1%) per pixel of 0.001x0.001 (~0.0124km2) for your specific area (rectangle of 100x100km2) would be around 800 points. If it is not a rectangle (referred to as 'region'), then we Readers expect the specific area.

126. Is this the way of citing Data? Check guidelines.

135-137. Rephrase, unclear. You mention MAGT as a 'proxy' for deformation? A control variable?

139. Why do you assume it is 'highly sensitive'? 

140. Increase? -- 

140. Rephrase, ¿What do you mean 'crucial factor' ? It is a variable you measure in order to compare different areas.

145-148 This sentence should be above, then explain about the variables you'll focus on in order to control deformation.

147. Where did you get ice-content data? From Cheng et al., 2019? If so, detail what and how you get. We need methodological information to REPLICATE this study for a broad audience.

153. Should web addresses appear here? Check guidelines for reference citations.

190. 'no deformation was assumed' what do you mean?

193. The points in Figure1 ?

199. What's the vertical resolution?

200. Are two points enough? Are these geodetic points? Did you establish the uncertainty and sensitivity analysis to assess the impact of control density on accuracy. Please check: https://doi.org/10.3390/rs13234800

282-285. This is a DISCUSSION/Limitations part.

286-287. What Results?

287-298. To Discussion...

345-371. Check what goes to Discussion and what remains as Result section.

374-385. If you talk about correlations you need to specify which corr. method you used and show results and p-values in a table for example.

386-399. You need to use the Figures/Tables to address the Result's statements you are writing. Any description or speculation about the Results goes to Discussion.

464. Improve this Figure, use just one distribution map or the combination of all three.

468. You are controlling deformation within the region (write specific area, or areal sum), not a regional assessment which requires extended geodetic controls.

471. You can remove the numbers, punctuation is enough for separate conclusion remarks.

487. 'We DID NOT detect...'

493-495. Rephrase this whole sentence and avoid citations.

The manuscript requires editing and some language revisions.

Reviewer 3 Report

This research presents high-quality analysis and visuals while outlining a meticulous and detailed method for evaluating permafrost deformation through Synthetic Aperture Radar (SAR) images. However, the section 3.1 SBAS-InSAR Processing (Line 172-217) , although thorough, is convoluted due to its continuous and undivided format. For enhanced clarity and understanding, it would be beneficial to categorize the information into distinct sections or paragraphs. Presenting this section in a more structured manner would greatly improve its comprehensibility, and thereby the overall impact of the research.

Firstly, a section devoted to the data and imagery used. Secondly, an in-depth explanation of the corrections made to the images should follow. The steps taken to generate a coregistered stack of SLC images and the subsequent generation of interferograms should be more detailed here. This would also be the appropriate section to explain the role of multilooking and phase unwrapping in the analysis.

Next, a detailed account of the georeferencing processes employed, along with the filters used to identify and extract outliers, would be useful. A clear explanation of how the displacement time series was geocoded into the WGS84 coordinate system and reprojected into the Albers area coordinate system would be beneficial to the reader.

Finally, a separate discussion should be dedicated to the impact of the earthquake on the deformation. This section would explain how the seismic event influenced the deformation time series and the measures taken to mitigate its impact.

Minor concerns:

Line 170... Table 2 is cited but I have not seen it in the text.

Line 182... 'the differential phase.' It starts with a lowercase letter.

Line 365... 'controlling water to determine the long-term subsidence rate of the ground surface.' It starts with a lowercase letter.

Round 2

Reviewer 1 Report

The manuscript has been improved and all my concerns have been adequately addressed. I have no more specific suggestions. 

Please carefully check the entire manuscript again to avoid any other mistakes since lots of changes have been made.

Author Response

Dear Reviewer,

Thank you very much for your recognition of our work. We have checked our manuscript again and improved the readability to some extent. The main changes were focused on the abstract, introduction and results sections.

Regards.

Reviewer 2 Report

Dear Authors,

You have met the comments in an adequate way. Please check your manuscript once again, improve readability and avoid any other mistakes such as using adjectives without a clear meaning.

Please clearly state where did you get the ground-ice content data and remember to explain how Zhou et al., analyses connect to your study.

Regards.

It should be checked once more as the Authors did change many things.
